# Task-and-Model-Aware Fractal-Consistency for Efficient LLM Reasoning

Ziqiu Luo [1]  Jianmin Liu [2]  Yukai Miao [3]  Li Chen [3]  Dan Li [2]

## Abstract

While self-consistency methods have emerged as a promising approach to enhance the correctness of large language model (LLM) outputs by aggregating multiple stochastic samples, they suffer from two critical limitations, resulting in high computation cost. First, they evaluate output consistency monolithically, failing to efficiently combine partially correct answers across multiple samples. Second, they use static stopping criteria that cannot adapt to varying task complexities and model capabilities, resulting in suboptimal computational efficiency. In this work, we present Task-and-Model-Aware Fractal-Consistency (TMAFC), a novel self-consistency framework that addresses these limitations through two key innovations: (1) Fractal-Consistency, which evaluates the output consistency at the granularity of output components to effectively combine partial correct answers across samples, and (2) Adaptive Stopping Criteria Calibration (ASCC), which dynamically adjusts sampling stopping criteria based on real-time assessment of both task difficulty and LLM capability. Through extensive experiments on diverse question-answering benchmarks, we demonstrate that TMAFC achieves superior efficiency-accuracy trade-offs, reducing sample cost by up to 55% while maintaining competitive accuracy compared to state-of-the-art baselines.

## 1. Introduction

Large language models (LLMs) have demonstrated remarkable capabilities across a wide range of tasks, including text, generation (Wei et al., 2022; Kojima et al., 2022), mathematical reasoning (Lewkowycz et al., 2022; Kang et al., 2025; Zhang et al., 2025), knowledge graph construction (Zhang & Soh, 2024; Sun et al., 2025; Bosselut et al., 2019), and code generation (Dong et al., 2025; Wang et al., 2025a; Nunez et al., 2024). However, their outputs often suffer from inconsistency and stochastic errors, limiting reliability in high-stakes applications.

To improve correctness, Self-Consistency (Wang et al., 2023; Kale & Nadadur, 2025; Bartsch et al., 2023; Prasad et al., 2025; Wan et al., 2025), has emerged as an effective technique, which aggregates multiple stochastic samples from an LLM and selects the most consistent output via majority voting. This approach significantly enhances robustness, particularly in reasoning tasks where a single sample may be error-prone. Since Self-Consistency requires a predefined number of samples (e.g., 50) (Wang et al., 2023), it struggles with high computational costs, especially when scaling to complex tasks.

Recent efforts have introduced early-stopping mechanisms to alleviate this burden. Early Stopping Self-Consistency (ESC) (Li et al., 2024b) adopts a segmented strategy, dividing the sampling process into windows and stopping early when answers within a window are all the same. Adaptive Consistency (AC) (Aggarwal et al., 2023) employs an incremental voting approach that dynamically terminates sampling once answers stabilize beyond a probabilistic threshold based on a probability distribution. However, existing consistency methods face two critical limitations:

- **The Monolithic Evaluation Bottleneck:** Current methods treat the entire output as an indivisible unit. This "all-or-nothing" evaluation is inherently inefficient. For structured tasks (e.g., JSON or code), an output may contain localized errors while being partially correct in other components. Monolithic stopping criteria fail to recognize these "converged" substructures, forcing redundant sampling of the entire response. Even in unstructured tasks, the latent structurality (e.g., independent constraints in logical subquestions) remains obscured, preventing fine-grained stopping and element-wise recombination across samples.

- **Lack of Task- and Model-Aware Adaptation:** Exist-

[1]Institute for Network Sciences and Cyberspace, Tsinghua University, Beijing, China [2]Department of Computer Science and Technology, Tsinghua University, Beijing, China [3]Zhongguancun Laboratory, Beijing, China. Correspondence to: Jianmin Liu <jianminl37@gmail.com>.

*Proceedings of the $43^{rd}$ International Conference on Machine Learning*, Seoul, South Korea. PMLR 306, 2026. Copyright 2026 by the author(s).

ing methods employ a static stopping criterion, failing to adapt to: 1) task complexity variations, resulting in either insufficient sampling for difficult tasks or wasteful oversampling for simple ones; 2) heterogeneous LLM capabilities, where less capable models typically demand more samples to achieve consensus. This results in suboptimal accuracy-cost trade-offs.

To address these limitations, we introduce Task-and-Model-Aware Fractal-Consistency (TMAFC), a novel framework that reimagines self-consistency through the lens of atomic decomposition and dynamic calibration. Our framework introduces two key innovations:

First, we propose Fractal-Consistency (FC). Moving beyond monolithic evaluation, FC decomposes LLM outputs into independently evaluable atomic units, such as key-value pairs in JSON or semantic "anchors" in unstructured reasoning. By tracking convergence at the component level, FC enables stable parts of an answer to exit the sampling loop early. This allows the framework to synthesize a final correct answer by recombining partial solutions from different samples, effectively achieving convergence in far fewer iterations than holistic methods.

Second, we propose Adaptive Stopping Criteria Calibration (ASCC), a mechanism that dynamically adjusts sampling termination. Unlike the static Beta stopping criteria in AC (Aggarwal et al., 2023) (where $\alpha_0 = 1, \beta_0 = 1$), ASCC calibrates the prior parameters $(\alpha_0, \beta_0)$ to match task complexity and LLM proficiency. Specifically, it categorizes tasks via an LLM-based classifier and determines optimal parameters through Maximum Likelihood Estimation (MLE) based on historical performance. This dual adaptation effectively reduces sampling costs while maintaining accuracy across diverse scenarios.

In summary, our key contributions are as follows:

- We propose TMAFC, a novel self-consistency framework for LLM reasoning, aiming to reduce computational costs while maintaining accuracy across diverse tasks and LLMs.

- We introduce Fractal-Consistency, which enables fine-grained component-wise termination and cross-sample element recombination for both structured and unstructured tasks.

- We develop ASCC, a robust architecture that uses MLE-based historical analysis and a difficulty classifier to dynamically calibrate sampling thresholds, adapting to task difficulties and LLM capabilities.

- We conduct extensive experiments using four LLMs across three structured and five unstructured tasks. Our results demonstrate that TMAFC significantly

outperforms the state-of-the-art method, Adaptive-Consistency. Specifically, TMAFC reduces sampling costs by up to 55% while improving accuracy by approximately 1.5% on structured tasks. For unstructured tasks, it cuts sampling costs by up to 38% with an about 0.8% increase in accuracy. We release the code implementation of TMAFC at: `https://github.com/MAX00101/TMAFC`.

## 2. Related Work

Self-Consistency (SC) (Wang et al., 2023), a decoding strategy, enhances LLM reasoning performance by sampling multiple outputs and aggregates outputs using a simple majority voting mechanism. While SC demonstrates improved reasoning accuracy (Huang et al., 2024), its performance strongly depends on the predetermined sample size. Notably, to maintain high accuracy, SC typically requires a large sample size (e.g., 40 as used in the original paper), resulting in high computational cost.

To mitigate computational cost, recent research has introduced early-stop mechanisms. (Li et al., 2024b) proposed Early-Stopping Self-consistency that divides the large sample size into multiple sequential small windows, terminating the process upon achieving answer uniformity within a window. Alternatively, (Aggarwal et al., 2023) introduced Adaptive-Consistency (AC) that adopts incremental sampling with a stopping criteria, terminating when a high-confidence majority as judged by the stopping criteria. Both ESC and AC adopt monolithic consistency evaluation at the level of the complete output, which may result in redundant computation, particularly for structured outputs, as they cannot combine partial correct elements from different samples. Moreover, neither method adapts sampling to task difficulty and LLM capabilities, resulting in insufficient sampling for challenging tasks or weaker LLMs and unnecessary oversampling for simple tasks or more capable models.

Despite Difficulty-Adaptive Self-Consistency (DSC) (Wang et al., 2025b) attempts to consider the influence of task difficulty on computational costs, it uses a static stopping criterion and depends on input batch questions without leveraging the target LLM's historical performance data. Moreover, its difficulty estimation relies on relative ranking across batched questions, which can be inaccurate and lack feasibility under heterogeneous task distributions and varying model capabilities, leading to unstable sampling allocation decisions. As a result, DSC still incurs high computational costs due to its inability to adaptively terminate sampling based on prior knowledge. Our TMAFC extends AC by addressing such limitations, aiming to improve computational efficiency while maintaining high accuracy across hybrid datasets in multiple domains.

## 3. Design

### 3.1. Preliminaries

**Beta Stopping Criteria.** Beta Stopping Criteria leverages the Beta distribution to model the probability distribution over unique output samples, a method proven to be highly efficient in AC (Aggarwal et al., 2023). Let $n$ denote the total number of samples generated by the LLM so far, with $m$ unique samples $[s_1, s_2, ..., s_m]$ and $[c1, c_2, ...c_m]$ be the counts of each element. Let $p_i = c_i/n$ be the normalized count. Among these, let $p_1$ be the majority element (i.e., the most frequent sample), $p_2$ be the next element. We assess the stability of $p_1$ over $p_2$ by computing the confidence level $\text{conf}_{p_1} = \psi(c_1, c_2)$ of $p_1$ using a beta distribution $\text{BETA}(\alpha_0 + c_1, \beta_0 + c_2)$:

$$\psi(c_1, c_2) = \frac{\int_0^{0.5} p_2^{\beta_0 + c_2 - 1}(1 - p_2)^{\alpha_0 + c_1 - 1}\, dp_2}{\text{BETA}(\alpha_0 + c_1, \beta_0 + c_2)} \qquad (1)$$

where $\alpha_0$ and $\beta_0$ are prior parameters. The sampling process terminates if $\text{conf}_{p_1}$ exceeds a predefined threshold $\tau$. The Beta stopping criterion offers a computationally efficient approximation to the Dirichlet stopping criteria (Aggarwal et al., 2023). Rather than evaluating the full distribution of probabilities over all unique samples as in Dirichlet stopping criteria, it focuses on establishing the dominance of the majority probability $p_1$ over the next largest probability $p_2$. As demonstrated in AC, this simplified approach maintains similar accuracy to the full Dirichlet formulation while achieving significantly faster inference speeds. Therefore, our TMAFC uses the Beta stopping criteria.

### 3.2. Overview

We propose Task-and-Model-Aware Fractal-Consistency (TMAFC), an innovative framework designed to optimize LLM reasoning by reducing computational costs while maintaining accuracy across diverse tasks and LLMs. Figure 1 shows an overview of TMAFC. We first propose Fractal-Consistency, an atomic consistency evaluation paradigm that assesses output consistency at the level of independent components of the output and applies a stopping criteria for each component, instead of traditional monolithic evaluation for the entire output using a single stopping criterion. To dynamically adapt to varying task complexities and LLM capabilities, we propose a novel Adaptive Stopping Criteria Calibration (ASCC) method that optimizes criteria parameters based on task-specific features and historical performance data from the target LLM. Specifically, for a given input question, TMAFC first adopts ASCC to derive the optimal criteria parameters $(\alpha_0, \beta_0)$, which are then used by Fractal-Consistency to generate the final answer.

### 3.3. Fractal Consistency

Existing self-consistency methods typically evaluate LLM outputs as monolithic, indivisible units. This paradigm suffers from two critical limitations: (1) **Coarse-grained Evaluation**: It cannot pinpoint local inaccuracies, forcing the re-sampling of an entire complex structure even when only a single component is erroneous. (2) **Redundant Computation**: It fails to recognize that different parts of a solution may converge at different rates. To address these, we propose **Fractal Consistency (FC)**, a paradigm that decomposes the output space into independently evaluable atomic units.

**Atomic Representation.** For any input question $Q$, we represent the target answer $A$ as a composite object $A = \{(a_1, v_1), (a_2, v_2), \ldots, (a_k, v_k)\}$, where each $(a_i, v_i)$ constitutes an *atomic component*—a pair representing a specific attribute or property and its corresponding value. The instantiation of these units depends on the task domain:

- **Structured Tasks:** For structured tasks like information extraction or code generation, these output units naturally map to explicit schema fields (e.g., specific keys in a JSON object or arguments in a function call). The set $\{a_i\}$ is predefined by the task schema.

- **Unstructured Tasks:** For unstructured reasoning tasks (e.g., mathematical or symbolic reasoning) that lack an explicit schema, directly applying component-wise consistency is challenging because the final output cannot be naturally decomposed into independent fields. Instead of physically decomposing the final answer, TMAFC introduces *anchors*, which are lightweight semantic logical sub-questions derived from the latent reasoning structure of the problem. Table 1 displays an example of generated anchors for a convoluted unstructured question. The purpose of these anchors is to constrain the reasoning path and reduce the variance of the LLM's answers to hard unstructural tasks, thus decreasing sampling cost without breaking apart the final output.

**Atomic-wise Convergence and Refinement.** FC tracks the convergence of each atomic unit independently. Formally, let $A^{(l)}$ denote the $l$-th sampled response. For each component $a_i$, we maintain a value collection $V_i = [v_i^{(1)}, \ldots, v_i^{(l)}]$. We identify the majority value $o_i \in V_i$ with frequency $c_i$, and the second-most frequent frequency $c_i'$.

The confidence $\text{conf}_{o_i} = \psi(c_i, c_i')$ is calculated via Eq. 1. Once $\text{conf}_{o_i} \geq \tau$, the component $a_i$ is marked as converged and moved to $A^*$. Crucially, converged components are excluded from the *pending set* $A_{pending}$. Subsequent sampling iterations use a refined prompt $P_j$ that provides the already-converged values as context and instructs the LLM

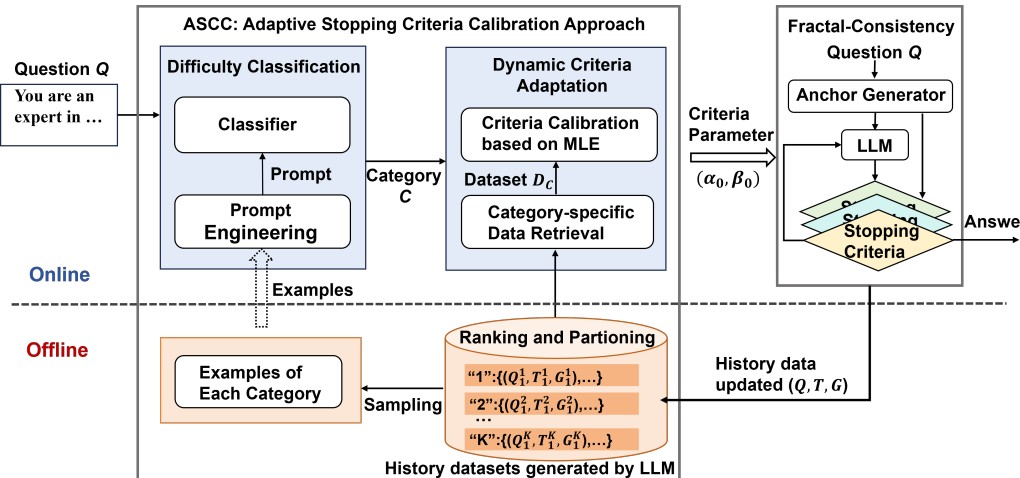

*Figure 1.* The overall architecture of our proposed TMAFC framework.

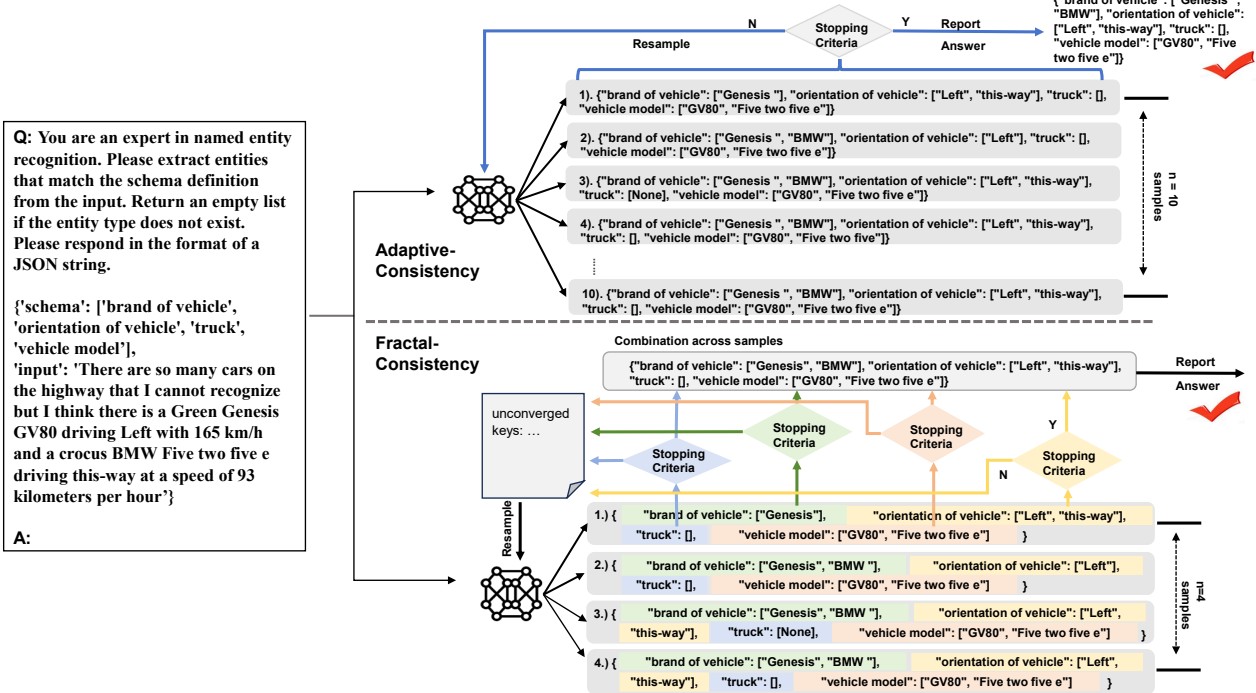

*Figure 2.* **Comparison between Adaptive Consistency (AC) and Fractal Consistency (FC).** AC evaluates consistency on the entire output as a monolithic unit. In contrast, FC decomposes the output into atomic components and applies stopping criteria independently to each component. In this example, the confidence threshold $\tau$ for the Beta stopping criterion is set to 0.9.

to focus exclusively on resolving the remaining uncertain components in $A_{pending}$. The process terminates when all atomic components achieve convergence.

For unstructured tasks, FC applies above consistency checking strictly on anchor answers except final answer to original question. After convergence of all anchors, TMAFC first scans the fully-generated samples and removes those that violate the converged anchors. TMAFC then performs a majority vote on the final answers of the remaining samples, finally outputting the ultimate choice. As illustrated in

Table 1, once the anchor consensus $A^* = [10, 120, 70]$ converges, trajectories inconsistent with the anchors are filtered out. The final answer *100* is then selected from the remaining coherent reasoning paths via standard answer voting. Note that if no sample aligns with the converged anchors, TMAFC defaults to standard majority voting across final answers.

Consequently, FC reduces redundant sampling by enabling early convergence on stable atomic components or logical anchors across samples. For instance, FC achieves a correct

*Table 1.* Illustration of FC for unstructured reasoning tasks.

**Question:** *A vehicle runs at 60 km/h and speeds up by 20 km/h every 2 hours. What is the average velocity before the 5th speed-up?*

| Anchors | Sample Round | | |
|---|---|---|---|
| | 1 | 2 | 3 |
| $A_1$: Total time before 5th speed-up? | 10 | 10 | 10 |
| $A_2$: Velocity after 3rd speed-up? | 120 | 100 | 120 |
| $A_3$: Avg. velocity before 2nd speed-up? | 70 | 60 | 70 |
| **Final Answer** | 100 | 90 | 100 |

JSON output in only 6 samples, compared to 13 for AC, as illustrated in Figure 2. We summarize FC contributing to both structural and unstructural tasks in Algorithm 1 and deliver more details about prompts of FC and anchor generation in Appendix A.

### 3.4. Adaptive Stopping Criteria Calibration Approach

As formalized in Eq. 1, the Beta stopping criteria relies on two key parameters of the Beta distribution: the prior parameters $\alpha_0$ and $\beta_0$. These parameters fundamentally determine the shape of the probability density function and consequently influence the confidence estimation. However, AC employs a static Beta stopping criteria based on a non-informative prior distribution with fixed parameters ($\alpha_0 = 1, \beta_0 = 1$) across all tasks and LLMs. This static stopping criteria fails to incorporate domain-specific prior knowledge and lacks adaptability to different difficulty tasks or LLMs with different capabilities.

In contrast, our Adaptive Stopping Criteria Calibration (ASCC) approach introduces a dynamic Beta stopping criteria based on an informative prior distribution with task- and model-aware parameters $(\alpha_0, \beta_0)$, allowing one to adapt to different task difficulties and LLM capabilities, thus improving convergence efficiency. As illustrated in Figure 1, ASCC consists of two key modules: Question Difficulty Classification and Dynamic Criteria Adaption.

**Question Difficulty Classification.** We introduce a Classifier designed to categorize questions into distinct difficulty levels. To ensure the classifier remains robust and adapts to the evolving capabilities of a target LLM on new or unseen tasks, we implement a dual-phase architecture. This architecture integrates real-time online classification with periodic offline refinement. During inference, for a given question, the classifier online categorizes this question into a specific difficulty level. Concurrently, an offline process periodically re-clusters all accumulated historical performance data into $K$ distinct difficulty categories.

*Offline Refinement.* The foundation of the Classifier is the offline refinement phase, which establishes and

---

**Algorithm 1** Fractal Consistency Algorithm

**Require:** Question $Q$, threshold $\tau$, max samples $N$
**Ensure:** Converged answer $A^*$
1: Initialize empty component sets: $A^* \leftarrow \emptyset$, $A_{pending} \leftarrow \{a_1, ..., a_k\}$
2: Initialize empty value collections: $\forall a_i, V_i \leftarrow \emptyset$
3: **if** Is_Unstructured($Q$) **then**
    $A_{pending} \leftarrow A_{pending} \cup$ LLM_AnchorGenerator($Q$)
4: **end if**
5: **for** $j \leftarrow 1$ to $N$ **do**
6:    Generate prompt $P_j$ targeting components $A_{pending}$
7:    Get LLM response: $A_j \leftarrow$ LLM($P_j$)
8:    **for** each component $a_i \in A_{pending}$ **do**
9:        Update value collection: $V_i \leftarrow V_i \cup \{v_i^j\}$
10:       Compute majority value $o_i$ and counts $(c_i, c'_i)$ from $V_i$
11:       Calculate confidence level: $conf_{o_i} \leftarrow \psi(c_i, c'_i)$ ▷ Using Eq. 1
12:       **if** $conf_{o_i} > \tau$ **then**
          Converge component: $A^* \leftarrow A^* \cup \{(a_i, o_i)\}$
          Remove from pending: $A_{pending} \leftarrow A_{pending} \setminus \{a_i\}$
13:       **end if**
14:    **end for**
15:    **if** $A_{pending} = \emptyset$ **then**
16:        **break** ▷ All components converged
17:    **end if**
18:    **if** Is_Unstructured($Q$) && $A_{pending} = \{a_1 \| a_1 \notin$ LLM_AnchorGenerator($Q$)$\}$ **then**
19:        **break** ▷ All components (anchors) converged
20:    **end if**
21: **end for**
22: **return** $A^*$

---

maintains an empirically-grounded definition of difficulty. This process centers on a historical performance database, $D = \{(Q_1, T_1, G_1), ..., (Q_j, T_j, G_j), ..., (Q_N, T_N, G_N)\}$, which chronicles the target LLM's success in answering a diverse set of questions. Each entry is a tuple where $Q_j$ is a historical question, $G_i$ represents the frequency of the majority answer, and $T_j$ is the sum of the counts for the majority answer and the second-most-frequent answer [1]. The database is initially populated through a comprehensive pre-sampling phase using the Adaptive-Consistency to ensure broad and representative coverage across various question types. In our experiments, we initializes the database with a pre-sampled set of 100 historical question-answering data, which we demonstrate in Figure 1 to provide statistically sufficient coverage for reliable online inference.

To define the difficulty categories, the Classifier first ranks all questions in the database by their required number of

---

[1]We focus only on the top two answers to remain consistent with the Beta stopping criteria.

samples ($T$) in ascending order. This ranking is based on the principle that questions requiring fewer samples to resolve are considered easier for the LLM. The ranked collection is then partitioned into $K$ equal-sized quantiles, where each quantile corresponds to a distinct difficulty level. This entire re-clustering process is executed periodically, incorporating all newly acquired query-response data to ensure that the difficulty definitions remain current and accurately reflect the LLM's performance. In our experiments, we perform re-clustering after every 50 new data. This interval ensures sufficient data coverage for reliable online inference while accommodating newly updated questions.

*Online Classification.* During online inference, the Classifier categorizes a new input question $Q$ in real time using the advanced semantic understanding of LLM itself through a dynamic quick-fire prompt strategy. The prompt consists of a system prompt concerning feature extraction and examples. We choose 7 features to denote the difficulty level of questions in system prompt, including: (1) text length of the question, (2) number of entities, (3) presence of conditional structures, (4) reasoning chain estimate, (5) clause estimation, (6) presence of vague or ambiguous terms, and (7) detection of large numbers. These quantified features are rigorously described to describe the difficulty of questions in different task domains. To construct the prompt, the Classifier first randomly samples $l$ representative historical questions from each of the $K$ difficulty categories established by the offline process. These examples $K \times l$, each paired with its known difficulty label, create a rich contextual prompt that effectively demonstrates the criteria for each category. Finally, LLM-based Classifier is prompted to analyze the new question $Q$ within the context of these examples and assign it to the most appropriate difficulty category. We provide the selected features and example prompt in Appendix B.

**Dynamic Criteria Adaptation.** After categorizing the input question into a specific classification $C$, we compute the optimal parameters $(\alpha_0, \beta_0)$ to determine the Beta stopping criteria for the question sampling based on the LLM's historical performance data in that category $C$ using MLE. Formally, let $D_C = \{(Q_1, T_1, G_1), (Q_2, T_2, G_2), ..., (Q_n, T_n, G_n)\}$ be the historical dataset for the target category $C$. We model the latent probability of the majority answer as $\theta_j$, follows a Beta prior distribution mentioned in Eq. 1 with unknown non-informative parameters $(\alpha_0, \beta_0)$:

$$\theta_j \sim \text{BETA}(\alpha_0, \beta_0)$$

The observed majority count $G_j$ follows a Binomial distribution conditioned on $T_j$ and $\theta_j$:

$$G_j \mid \theta_j \sim \text{Binomial}(T_j, \theta_j)$$

We can then calculate the probability of joint distribution of $\theta_j$ and $G_j$ by multiplying these two probability density functions and then get:

$$P(G_j, \theta_j) = \binom{T_j}{G_j} \frac{\theta_j^{\alpha_0 + G_j - 1}(1 - \theta_j)^{\beta_0 + T_j - G_j - 1}}{B(\alpha_0, \beta_0)} \quad (2)$$

The $B(\cdot, \cdot)$ denotes beta function. By marginalizing out the latent variable $\theta_j$, the marginal likelihood of observing $G_j$ successes out of $T_j$ trials follows a Beta-binomial distribution:

$$P(G_j \mid T_j, \alpha_0, \beta_0) = \binom{T_j}{G_j} \frac{B(\alpha_0 + G_j, \beta_0 + T_j - G_j)}{B(\alpha_0, \beta_0)} \quad (3)$$

Since the binomial coefficient $\binom{T_j}{G_j}$ is a constant with respect to $\alpha_0$ and $\beta_0$, we drop it from the objective function, leading to the likelihood function for the Beta distribution given the observed data $D$:

$$\mathcal{L}(\alpha_0, \beta_0 \mid D_C) = \prod_{j=1}^{n} \frac{B(\alpha_0 + G_j, \beta_0 + T_j - G_j)}{B(\alpha_0, \beta_0)} \quad (4)$$

We calculate the optimal $\alpha_0$ and $\beta_0$ by maximizing the following log-likelihood:

$$\ln \mathcal{L}(\alpha_0, \beta_0 \mid D_C) = \sum_{j=1}^{n} \ln \frac{B(\alpha_0 + G_j, \beta_0 + T_j - G_j)}{B(\alpha_0, \beta_0)}$$
$$\text{subject to:} \quad \varepsilon_{\min} \le \alpha_0, \beta_0 \le \varepsilon_{\max} \quad (5)$$

where the bounds $[\varepsilon_{\min}, \varepsilon_{\max}]$ prevent parameter overfitting to historical data artifacts. We solve the above log-likelihood maximization problem using L-BFGS-B (Byrd et al., 1995), a gradient-based method particularly suited for box-constrained problems, which efficiently minimizes the negative log-likelihood (i.e., $-\ln \mathcal{L}(\alpha_0, \beta_0 \mid D_C)$). We summarize ASCC in Algorithm 2.

## 4. Experiments

We conduct a comprehensive evaluation of TMAFC across 8 diverse benchmark datasets, encompassing both structured and unstructured output formats, and testing with 4 distinct LLMs. Based on these extensive experiments, we present a systematic analysis of TMAFC's performance variations across different tasks and model capabilities.

### 4.1. Experimental Setup

**Datasets, Models and Baselines.** To comprehensively evaluate TMAFC, we conduct experiments across both structured and unstructured output tasks. For structured tasks, we focus on knowledge graph attribute extraction across three

**Algorithm 2** Adaptive Stopping Criteria Calibration

---

**Require:** Current question $Q$, the LLM's historical performance dataset $D$

**Ensure:** Optimal parameters $(\alpha_0^*, \beta_0^*)$

    **Module 1: Question Difficulty Classification**

    $C \leftarrow \text{Classifier}(Q)$             ▷ Categorize question

    **Module 2: Dynamic Criteria Adaptation**

    Retrieve category-specific history data $D_C \subseteq D$

    Initialize $\alpha_0 \leftarrow 1$, $\beta_0 \leftarrow 1$

    Solve a MLE problem: $\max\limits_{\alpha_0, \beta_0} \ln \mathcal{L}(\alpha_0, \beta_0 \mid D_C)$ using

    L-BFGS-B optimizer by minimizing $-\ln \mathcal{L}(\alpha_0, \beta_0 \mid D_C)$ using Eq. 5

    **return** $(\alpha_0^*, \beta_0^*)$

---

distinct categories from IEPile (Gui et al., 2024): Name Entity Extraction (NER), Relation Extraction (IE), Event Extraction (EE), aggregating about 1000 questions. For unstructured tasks, we utilize standard benchmarks including mathematical reasoning (GSM-hard (Li et al., 2024a), Math-P-hard (Huang et al., 2025)) and symbolic reasoning tasks (Penguins (Gorman et al., 2014), Disambiguation_QA (Hannan et al., 2020), with around 2000 questions. For fair comparison with baseline methods in previous state-of-the-art with classic and latest models, we evaluate TMAFC on four LLMs used in AC and ESC: GPT-4 (OpenAI et al., 2023), GPT-3.5-TURBO (OpenAI, 2023), QWEN3-14B (Yang et al., 2025), and LLAMA3-8B-Instruct (Meta, 2024). We compare TMAFC with two baseline methods: AC (Aggarwal et al., 2023) and ESC (Li et al., 2024b), which are two strong baselines for cost-efficient self-consistency methods, we reproduce both methods following their original implementations to ensure a fair comparison.

**Implementation details.** We deploy open-sourced LLMs locally on 2 A6000 NVIDIA GPUs and utilize official APIs of close-sourced LLMs for sampling inferences. The inference temperature for all LLMs is 0.7. We set the number of difficulty levels $K$ to 3 (reasonableness detailed in Appendix C, sampled examples $l$ for each classification to 2, and confidence threshold $\tau$ for consistency assessment to 0.95. The bounds for the adaptive stopping criteria are set to $[\varepsilon_{\min}, \varepsilon_{\max}] = [0.01, 1]$ for both parameters $\alpha_0$ and $\beta_0$ optimization.

### 4.2. Main Results

In this experiment, we utilize the reasoning LLM itself as LLM-based Anchor Generator and Difficulty Classifier to avoid fairness concerns that TMAFC relies on stronger proprietary model to conduct sampling process. We further manually evaluate these two auxiliary reasoning modules on a sampled set of 300 tasks. The accuracy of anchor generation ranges from 55% to 93% across different unstructured reasoning tasks, while the difficulty classifier achieves classification accuracy ranging from 68% to 84% under different task domains and model settings.

We first evaluate the performance of different methods across structured and unstructured tasks using GPT-4, with sample limits set to 50 for NER and unstructured tasks, and 100 for RE and EE tasks. On average, the ASCC mechanism categorizes structured questions into three difficulty levels as follows: 38.4% as *Easy*, 37.2% as *Middle*, and 24.4% as *Hard*; and unstructured tasks are split into 42.4% as *Easy*, 38.5% as *Middle*, 19.6% as *Hard*. As shown in Table 2 and Table 3, TMAFC achieves superior performance across all structured tasks and unstructured tasks, reducing the sampling cost by approximately 55% and 38% respectively, while slightly improving accuracy by 0.4%–1.5% compared to AC.

To further evaluate the individual contributions of FC and ASCC, we perform ablation studies by comparing against: (1) TMAFC without ASCC (w/o ASCC) and (2) TMAFC without FC (w/o FC). As shown in Table 2 and Table 3, the ablation study reveals two key findings: (1) FC-only variant (w/o ASCC) achieves substantial sampling cost reduction (above 45%) with slight accuracy improvements. (2) The complete TMAFC (with both FC and ASCC) achieves further sampling reductions (up to 55%), with only marginal (less than 0.2%) accuracy trade-offs, demonstrating the complementary nature of these two mechanisms. We find that the unstructured experiment implies consistent consequences with structured evaluation, which verify the validity of our anchor generator desination.

Next, we further evaluate TMAFC and AC across hybrid structured and unstructured tasks using different LLMs, with a fixed sample limit of 50 per task, to further investigate the proposed TMAFC's end-to-end performance across different models and task domains. The classifier partitions the hybrid dataset into three difficulty levels according to different distributions for each model, which is shown in the task difficulty level column of Table 4. For GPT-4, due to its strong reasoning capability, the majority of questions are classified as *Easy*. In contrast, for LLAMA3-8B-Instruct, most questions fall into the *Middle* and *Hard* categories, reflecting its relatively limited problem-solving ability.

Unlike AC uses a fixed non-informative parameters ($\alpha_0 = 1, \beta_0 = 1$), TMAFC adaptively optimizes $\alpha_0$ and $\beta_0$ parameters according to task difficulty and LLM capabilities. From Table 4, we observe that, TMAFC increases $\beta_0$ when encountering either more challenging tasks or weaker LLMs, thereby automatically acquiring additional samples to ensure reliable consensus. As a result, TMAFC achieves a substantial reduction (about 32%) in sample cost across all LLM & task combinations, while maintaining comparable or nearly identical answer accuracy, compared with AC.

*Table 2.* **GPT-4** Performance of different methods across structured tasks

| Method | Task Domain | | | | | |
|--------|-------------|---|---|---|---|---|
| | NER | | RE | | EE | |
| | Sampling Cost | Accuracy | Sampling Cost | Accuracy | Sampling Cost | Accuracy |
| SC | 50 | 85.4% | 100 | 66.5% | 100 | 42.1% |
| ESC | 42.31 (-15.8%) | 85.4% (+0.0%) | 65.8(-34.2%) | 66.5% (+0.0%) | 77.2(-28.8%) | 42.1% (+0.0%) |
| AC | 36.81 (-26.38%) | 85.4% (+0.0%) | 45.34 (-54.66%) | 66.7% (+0.2%) | 56.83 (-43.17%) | 41.8% (-0.3%) |
| TMAFC | **16.44(-67.12%)** | **85.8% (+0.4%)** | **21.48(-78.52%)** | 67.2% (+0.7%) | **32.25(-67.75%)** | 43.6% (+1.5%) |
| w/o ASCC | 19.32 (-61.36%) | 85.8% (+0.4%) | 24.46 (-75.54%) | **67.4% (+0.9%)** | 35.71 (-64.29%) | **43.7% (+1.6%)** |
| w/o FC | 25.56(-48.88%) | 85.4% (+0.0%) | 29.44(-70.56%) | 66.7% (+0.2%) | 36.65(-63.35%) | 41.8% (-0.3%) |

*Table 3.* **GPT-4** Performance of different methods across unstructured tasks

| Method | Task Domain | | | | | |
|--------|-------------|---|---|---|---|---|
| | Mathematical Reasoning | | Math-Reasoning (Hard) | | Symbolic Reasoning | |
| | Sampling Cost | Accuracy | Sampling Cost | Accuracy | Sampling Cost | Accuracy |
| SC | 50 | 79.4% | 50 | 68.1% | 50 | 84.3% |
| ESC | 44.9 (-10.2%) | 78.7% (-0.7%) | 44.72 (-10.56%) | 66.3% (-1.8%) | 43.7 (-12.6%) | 83.1% (-1.2%) |
| AC | 38.6 (-22.8%) | 79.4% (+0.0%) | 42.4 (-15.2%) | 67.6% (-0.5%) | 35.34 (-29.32%) | 84.3% (+0.0%) |
| TMAFC | **21.24 (-57.52%)** | **80.2% (+0.8%)** | **30.63 (-38.74%)** | **68.5% (+0.4%)** | **19.77 (-60.46%)** | **84.8% (+0.5%)** |
| w/o ASCC | 26.34 (-47.32%) | 80.2% (+0.8%) | 33.1 (-33.8%) | 68.1% (+0.0%) | 22.61 (-54.78%) | 84.8% (+0.5%) |
| w/o FC | 28.52 (-42.96%) | 79.4% (+0.0%) | 39.82 (-20.56%) | 68.1% (+0.0%) | 23.75 (-52.5%) | 84.3% (+0.0%) |

*Table 4.* Performance comparison across different LLMs and hybrid **unstructured & structured** tasks of varied difficulty levels

| Models | Task Difficulty Level | $(\alpha_0, \beta_0)$ | | Sampling Cost | | Accuracy | |
|--------|----------------------|------|------|------|------|------|------|
| | | AC | TMAFC | AC | TMAFC | AC | TMAFC |
| GPT-4 | Easy (86.72%) | (1,1) | (1, 0.1) | 5.23 | 3.11 (-40.5%) | 84.7% | 84.7% (-0.0%) |
| | Middle (7.27%) | (1,1) | (1, 0.25) | 9.08 | 5.04 (-44.5%) | 65.47% | 65.93% (+0.46%) |
| | Hard (6.01%) | (1,1) | (1, 0.58) | 22.06 | 18.58(-15.8%) | 51.06% | 51.81% (+0.75%) |
| | Average | - | - | 8.61 | **4.14 (-52.3%)** | 81.27% | **81.38% (+0.11%)** |
| GPT-3.5 Turbo | Easy (38.01%) | (1,1) | (1, 0.33) | 6.69 | 4.15 (-38.0%) | 71.38% | 71.38% (-0.0%) |
| | Middle (29.62%) | (1,1) | (1, 0.55) | 15.66 | 11.73 (-25.1%) | 61.79% | 62.32% (+0.53%) |
| | Hard (32.36%) | (1,1) | (1, 0.84) | 34.8 | 30.1 (-13.5%) | 59.2% | 60.68%(+1.48%) |
| | Average | - | - | 18.44 | **14.79 (-19.7%)** | 64.59% | **65.24% (+0.65%)** |
| QWEN3-14B | Easy (41.2%) | (1,1) | (1, 0.22) | 6.84 | 3.92 (-42.7%) | 36.3% | 36.3% (-0.0%) |
| | Middle (35.8%) | (1,1) | (1, 0.53) | 16.6 | 13.2 (-20.4%) | 26.6% | 27.8% (+1.2%) |
| | Hard (23.0%) | (1,1) | (1, 0.7) | 36.31 | 28.92 (-20.4%) | 23.3% | 23.8% (+0.5%) |
| | Average | - | - | 17.09 | **12.98 (-24.1%)** | 29.8% | **30.34% (+0.54%)** |
| LLAMA3-8B-Instruct | Easy (15.2%) | (1,1) | (1, 0.43) | 7.52 | 5.14 (-31.6%) | 15.3% | 15.3% (-0.0%) |
| | Middle (45.1%) | (1,1) | (1, 0.75) | 18.43 | 16.62 (-9.8%) | 8.62% | 8.98% (+0.36%) |
| | Hard (39.7%) | (1,1) | (1, 0.95) | 38.3 | 35.6 (-7.0%) | 7.3% | 7.3% (-0.0%) |
| | Average | - | - | 24.66 | **22.4 (-9.1%)** | 9.1% | **9.26% (+0.16%)** |

These results demonstrate TMAFC's effectiveness in achieving optimal accuracy-efficiency trade-offs and its strong generality as an atomic consistency method and adaptive stopping criterion for LLM inference tasks.

### 4.3. Additional Overhead and Robustness

Beyond sampling efficiency, we analyze the overhead and robustness of TMAFC 's auxiliary modules, including end-to-end cost, anchor reliability, and difficulty classification stability.

**End-to-End Cost and Latency Analysis.** TMAFC introduces two auxiliary LLM modules for unstructured tasks: an anchor generator for FC and a difficulty classifier for ASCC. Both modules are executed in batched mode to amortize fixed prompt overhead across queries. For each query, we account for all consumed tokens, including reasoning sampling, anchor generation, and difficulty classification.

Figure 3 breaks down the end-to-end token and latency overhead of TMAFC. For unstructured tasks, auxiliary modules account for only $\sim 24\%$ of total tokens ($\sim 15\%$ anchors and $\sim 9\%$ classifier), while overall token consumption is still reduced by $\sim 44\%$ compared with AC due to fewer sampling iterations. Also, the additional latency overhead remains lightweight ($\sim 28\%$ combined), yet end-to-end latency is reduced by more than $50\%$ relative to AC. For structured tasks, classifier introduces only $\sim 8\%$ extra token consumption and $\sim 6\%$ latency, while reduces more overall cost by $\sim 52\%$ in token and $\sim 59\%$ in latency respectively, since anchor generation is removed. These results indicate that sampling dominates overall inference cost, while the auxiliary modules introduce only affordable overhead that is consistently outweighed by adaptive sampling reduction.

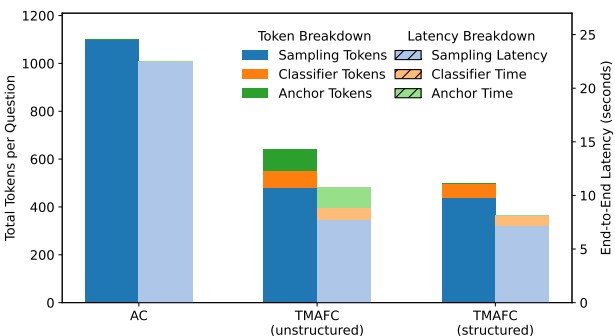

*Figure 3.* **End-to-end token and latency cost.** Total cost is decomposed into sampling, anchor generation, and difficulty classification.

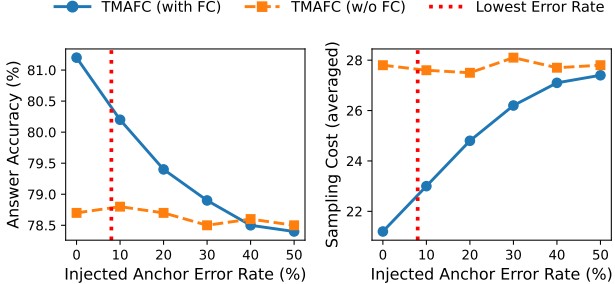

*Figure 4.* **Sensitivity to anchor quality.** We inject increasing levels of synthetic noise into oracle anchors.

**Resilience to Imperfect Anchor Generation.** On a manually annotated subset of approximately 300 unstructured problems, the anchor generator achieves best accuracy of 92.7% with GPT-4 and worst of 55.7% with llama3-8B-Instruct, while most errors arise from over-segmentation or error value deviations rather than generating logically simpler sub-questions. Such errors were amplified in weaker LLMs like Llama3-8B-Instruct.

To further assess robustness, we inject synthetic noise into oracle anchors by randomly dropping, perturbing, or misassigning a fraction of anchors, simulating error rates from 0% to 50%. Figure 4 shows that when anchors are accurate, FC yields both higher answer accuracy and lower sampling cost than an ASCC-only method. As anchor quality degrades, performance decreases smoothly without abrupt failures: answer accuracy gradually approaches that of the ASCC-only variant, while sampling cost increases and eventually converges to the same baseline. These results indicate that anchor errors primarily reduce efficiency rather than causing systematic correctness failures, enabling a safe fallback to ASCC-only inference when anchor confidence is low.

**Impact of Difficulty Classification.** We further evaluate robustness to classifier variability on a cross-domain dataset of 300 questions. LLM-based classifiers (tested LLMs in the main results and other mainstream LLMs like DeepSeek-

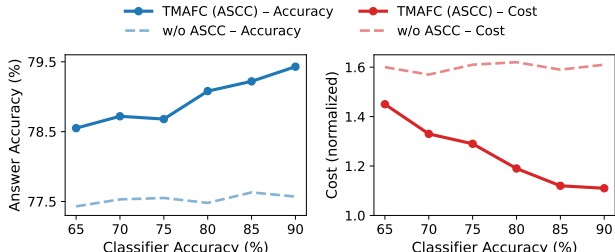

*Figure 5.* **Sensitivity to classification noise under realistic accuracy ranges.** We simulate GPT-4-based classifier accuracy from 65% to 90% via noise injection into oracle labels.

V3 (AI, 2024)) exhibit accuracy fluctuations in the range of 66%–84% across our repeated measurements. In Figure 5, sensitivity analysis with injected noise shows that ASCC remains stable when classifier accuracy lies between 65% and 90%, both performances on answer accuracy and end-to-end cost of TMAFC consistently outperforms the w/o-ASCC variant across the full range, demonstrating the robustness of LLM-based classifier embedded in ASCC stage. Furthermore, our self-trained lightweight classifiers such as XGBoost (Chen & Guestrin, 2016) and DistilBERT (Sanh et al., 2019) fall outside this robustness regime (detailed in Appendix C) and lead to inferior end-to-end performance, indicating that difficulty classification for ASCC requires strong semantic and logical modeling capacity, and that LLM-based classifiers constitute the most effective solution for cross-domain deployment.

## 5. Conclusion

In this work, we present TMAFC, a novel self-consistency framework for efficient LLM reasoning. Unlike conventional self-consistency approaches that evaluate consistency at the monolithic output level, our proposed Fractal-Consistency mechanism operates at the atomic output level, enabling the aggregation of partial correctness across samples. Furthermore, we introduce ASCC to dynamically determine optimal sampling termination parameters based on real-time assessment of task difficulty and model capabilities, overcoming the limitations of static stopping criteria employed by existing methods. Extensive experiments demonstrate that TMAFC achieves substantial computational savings while maintaining competitive accuracy.

## Acknowledgements

This work was supported by the Beijing Outstanding Young Scientist Program (No. JWZQ20240101008).

## Impact Statement

This work improves the efficiency of large language model reasoning by reducing unnecessary sampling while main-

taining answer quality. We do not identify specific societal risks beyond the broader limitations of LLMs, including potential reasoning errors and hallucinations.

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

---

**Fractal Consistency Prompt Example**

You are an expert in named entity recognition. Please extract entities that match the schema definition from the input. Return an empty list if the entity type does not exist. Please respond in the format of a JSON string.

For example:
{"schema": ["biomedical", "engineering feature", "concept or principle", "material", "machine or equipment", "process parameter"],
"input": "A 3D numerical model is developed to study the flow mechanism with rotation nozzle at the corner under various conditions during the extrusion and deposition process ; Material rheological properties have little effects on material distribution ratio at corners , while process parameters affect material distribution ratio significantly ; Increasing corner radius and relative nozzle travel speed while decreasing nozzle aspect ratio are beneficial to suppressing uneven mass distribution at corners ; When conducting corner printing with rotational rectangular nozzle , a greater amount of material is deposited inside the filament and hence tearing and skewing will occur on the surface of the printed filament.}
"output": {"biomedical": [], "engineering feature": ["aspect ratio"], "concept or principle": ["3D", "model", "mechanism", "properties", "distribution", "process parameters", "surface"], "material": ["Material", "material", "filament"], "machine or equipment": ["nozzle"], "process parameter": []}

Question:
{'schema': [Hours, Price, Location, Cuisine. ],
'input': 'is there an irish place with a view nearby'}

Output:

---

*Figure 6.* **Fractal Consistency Prompt example for a structured NER task.** As ASCC confirms variables with sufficient confidence, they are removed from the schema list, allowing subsequent samples to focus only on unresolved components.

## A. Details of Fractal Consistency and Anchor Generation

This appendix provides additional details on Fractal Consistency (FC), including its execution process, prompt construction, and its applicability to both structured and unstructured tasks. While several figures illustrate FC using structured prediction (e.g., NER) for clarity, the underlying mechanism is task-agnostic and applies equally to unstructured reasoning via anchor-based decomposition.

### A.1. Fractal Consistency Prompt Example

Algorithm 1 describes the FC procedure. At each iteration, the prompt is dynamically updated to include only unresolved components. Variables whose confidence already exceeds the ASCC threshold are removed from the prompt, while unconverged variables remain.

Figure 6 shows a Fractal Consistency Prompt example for a structured NER task. The *Question* section contains an explicit schema that defines the set of variables to be extracted. As sampling progresses and ASCC confirms certain variables with sufficient confidence, the schema list is progressively reduced. This process continues until either all variables converge or the maximum number ofsampling iterations is reached. Figure 7 further illustrates the Fractal Consistency Prompt for unstructured questions equipped with anchors. LLM conduct reasoning on these anchors and original question in every sample turn.

Although the example uses NER for illustration, the Fractal Consistency Prompt mechanism does not depend on the task being structured. The key requirement is the existence of identifiable atomic components whose outputs can be evaluated independently. By focusing the model's capacity on unresolved variables, FC enables the model to resolve simple components early and concentrate computation on harder components, leading to lower sampling cost and improved accuracy.

### A.2. Anchor Generation for Unstructured Tasks

For unstructured tasks where outputs cannot be naturally expressed as a fixed schema, TMAFC introduces an anchor generator to enable Fractal Consistency. Given a problem, the anchor generator produces a small set of simpler and independent logical sub-questions, referred to as anchors, derived from the original problem itself.

---

**Fractal Consistency Prompt Example   (Unstructured)**

You are an expert in reasoning. Given a question and a set of anchors, you must answer each anchor question and the original question separately, and return all results in the specified JSON format.

Question:
A motel runs at 60km/h and speeds up after every 2 hours, what's the average velocity of motel before the 5th speed-up?

Anchors:
- How many hours does the motel run in total before its 5th speed-up?
- What's the velocity after 3rd speed-up?
- What's the average velocity before 2nd speed-up?

Output:
{
  Anchor_1: 10,
  Anchor_2: 120,
  Anchor_3: 70,
  Final_Answer: 100
}

Questions:
<Target_Question_Description>

Anchors:
<LLM_Generated_Anchors>

Output:

*Figure 7.* **Fractal Consistency Prompt example for unstructured task.** LLM simultaneously answer the anchors and original question, but only conduct ASCC process and early stop with anchors.

These anchors correspond to semantic invariants, intermediate logical properties, or decomposable sub-conditions that a correct reasoning trajectory is expected to satisfy. Unlike explicit output fields in structured tasks, anchors do not directly decompose the final answer. Instead, they project the original reasoning problem into a lower-dimensional space of independently checkable logical sub-questions, allowing simpler semantic properties to converge earlier than the full reasoning process.

Anchors serve as atomic consistency units for FC in unstructured settings. Each anchor is evaluated independently for consistency across samples. Once an anchor reaches sufficient confidence, it is removed from subsequent prompts, mirroring the behavior of resolved variables in structured tasks.

Importantly, FC does not reconstruct the final answer from converged anchors. Instead, anchor consistency is applied strictly at the level of logical sub-questions. After anchor convergence, TMAFC scans the fully-generated reasoning samples and removes those whose anchor answers violate the converged anchor consensus. The final answer is then selected holistically from the remaining coherent reasoning paths through majority voting.

This design allows FC to preserve complete and coherent reasoning trajectories, rather than constructing answers by stitching together partial intermediate states.

Figure 8 shows the anchor generator prompt used for unstructured tasks. The prompt explicitly constrains anchors to be concrete, independently verifiable, and grounded in the original problem statement, while forbidding explanations or full reasoning chains. This design ensures that anchors function as atomic consistency checks rather than latent reasoning steps.

By converting unstructured reasoning into a set of independently evaluable logical sub-questions, FC unifies structured and unstructured tasks under the same consistency framework. In both cases, the stopping criterion operates on atomic components, enabling adaptive allocation of sampling effort and graceful degradation when certain components are ambiguous or noisy.

---

**Anchor Generator Prompt**

You are an Anchor Generator.
Your goal is NOT to solve the problem.
Given a problem and a candidate answer, you must generate a small set of independent, checkable anchors that must hold if the candidate answer is correct.
An anchor is a necessary condition, not an explanation or reasoning step.
You must follow these rules strictly:
Do NOT explain reasoning.
Do NOT restate the answer.
Each anchor must be verifiable in isolation.
Anchors should be short, concrete, and grounded in the problem statement.
Anchors may refer to intermediate quantities, relations, constraints, or required properties.
If an anchor cannot be verified due to ambiguity, still output it; uncertainty is allowed.
You are allowed to generate anchors that may later turn out to be false — correctness will be assessed by consistency, not by you.

Problem:
{PROBLEM_TEXT}

Candidate Answer:
{CANDIDATE_ANSWER}

Generate at most {K} anchors.

Output format (STRICT):

ANCHOR i:
type: <one of [numeric, relational, logical, semantic, coverage, consistency]>
description: <one short sentence describing a necessary condition>
check: <a concrete checkable statement or pseudo-assertion>

Notes:
- The "check" field should be something that could be evaluated as True / False / Unknown.
- Use values, entities, or relations explicitly mentioned or implied in the problem.
- Do NOT include explanations or derivations.

*Figure 8.* **Anchor generator prompt example for unstructured tasks.** The generator produces a bounded set of independent, checkable anchors that serve as atomic units for Fractal Consistency.

## B. Prompt Design for the TMAFC Difficulty Classifier

This appendix describes the prompt design of the LLM-based difficulty classifier used in TMAFC. The classifier prompt is composed of two complementary parts: a fixed system prompt that encodes general difficulty assessment principles, and a small set of few-shot examples that provide concrete calibration signals. These two components play distinct roles and operate on different time scales.

### B.1. Offline System Prompt for Difficulty Assessment

Figure 9 shows the system prompt used by the difficulty classifier. This prompt is designed offline and remains fixed during deployment. Its role is to guide the LLM to assess question difficulty based on general semantic and logical characteristics rather than domain-specific surface cues.

The system prompt specifies a set of difficulty-related features, including linguistic complexity, logical structure, compositionality, ambiguity, and the depth of reasoning required. Together, these features capture both textual and semantic aspects of problem difficulty and are intentionally defined in a domain-agnostic manner. By emphasizing sentence-level semantics and logical patterns, the system prompt promotes cross-domain generality and enables the classifier to operate consistently across heterogeneous tasks.

Because the system prompt is fixed and shared across all queries, its cost is amortized via batched inference. More

importantly, keeping this component offline ensures that difficulty assessment criteria remain stable over time, preventing unintended drift in classifier behavior.

---

**Classification System Prompt**

You are a question difficulty evaluator. Your task is to assign a difficulty level to a question based on its text, using the following scale:
0 (Easy), 1 (Medium), 2 (Hard).
Your evaluation is based on a holistic judgment considering these criteria:

1. Text length of the question.
2. Number of entities (people, objects, numbers, etc.) in the question.
3. Presence of conditional structures (e.g., "if... then...").
4. Reasoning chain estimate – are there multiple dependency steps or jumps required?
5. Clause estimation – based on commas, conjunctions, or specific syntactic cues.
6. Presence of vague or ambiguous terms that could cause confusion.
7. If the question involves computation, determine whether the numbers used are large (e.g., over one million), which may affect complexity.

You must output only the difficulty level (0, 1, or 2) — no other text.

Examples:
…
Question:
…

---

*Figure 9.* **System prompt of the difficulty classifier.** The prompt encodes general, domain-agnostic features for assessing question difficulty and remains fixed during deployment.

### B.2. Online Few-Shot Examples for Calibration

Figure 10 shows the few-shot examples included in the classifier prompt. Unlike the system prompt, these examples are maintained online and can be updated periodically based on newly observed data.

The few-shot examples provide concrete reference points that calibrate the classifier's output scale and decision boundaries. They reflect representative questions with labeled difficulty levels drawn from historical workloads. By updating these examples over time, TMAFC can adapt to shifts in task distributions while preserving the underlying difficulty assessment principles encoded in the system prompt.

This separation between a fixed system prompt and an adaptive example set allows TMAFC to balance stability and flexibility. The system prompt defines *what* constitutes difficulty in a general sense, while the few-shot examples adjust *how* these criteria are instantiated under evolving workloads. Empirically, this design contributes to the robustness of the LLM-based classifier observed in Section 4.3 and Appendix C.

## C. Details of Classifier Evaluation

### C.1. Experimental Setup

We evaluate difficulty classification for ASCC using a mini-benchmark composed of 300 questions spanning three task families: (i) structured named-entity recognition (NER), (ii) mathematical reasoning, and (iii) logical reasoning. For each task family, we construct balanced evaluation sets and sweep the number of difficulty levels $K \in \{2, \ldots, 7\}$. We compare LLM-based classifiers (GPT-3.5-Turbo, GPT-4, and DeepSeek-V3) with lightweight baselines (XGBoost and DistilBERT), as well as an oracle classifier. For each configuration, we report end-to-end answer accuracy and sampling cost under TMAFC with ASCC enabled.

### C.2. Effect of Difficulty Granularity $K$

Figure 11 reports the effect of varying $K$ across classifiers and task domains. Across all LLM-based classifiers and all three task families, $K{=}3$ consistently achieves the best end-to-end trade-off, yielding both the highest answer accuracy and the lowest sampling cost. More detailed analysis reveals distinct failure modes when deviating from this setting.

When $K{=}2$, the difficulty granularity is too coarse. Questions with substantially different intrinsic difficulty are merged

---

**Classification Examples**

Examples:

Input: Janet's ducks lay 16 eggs per day. She eats three for breakfast every morning and bakes muffins for her friends every day with 4933828. She sells the remainder at the farmers' market daily for $2 per fresh duck egg. How much in dollars does she make every day at the farmers' market?
Output: 1

Input: A robe takes 2287720 bolts of blue fiber and half that much white fiber. How many bolts in total does it take?
Output: 0

Input: Kylar went to the store to buy glasses for his new apartment. One glass costs $5, but every second glass costs only 60% of the price. Kylar wants to buy 5364765 glasses. How much does he need to pay for them?
Output: 0

Input: Toulouse has twice as many sheep as Charleston. Charleston has 4 times as many sheep as Seattle. How many sheep do Toulouse, Charleston, and Seattle have together if Seattle has 6314246 sheep?
Output: 2

Input: A merchant wants to make a choice of purchase between 2 purchase plans: jewelry worth $5,000 or electronic gadgets worth $8,000. His financial advisor speculates that the jewelry market will go up 2.5% while the electronic gadgets market will rise 1.2% within the same month. If the merchant is looking to maximize profit at the end of this month by making a choice, how much profit would this be?
Output: 1

Input: Ariadne has a shop selling hats of two different colors, red and green. Her sales from red hats were $2437532 in a particular month, half the total amount she earned from selling green hats. Calculate the total amount she made in two months if in the second month her sales were 3/4 of the total sales of the first month.
Output: 2

---

*Figure 10.* **Few-shot classification examples.** The example set is updated online to reflect current workloads and calibrate the classifier's difficulty predictions.

into the same bucket, leading to unstable estimation of ASCC prior hyperparameters. As a result, the derived Beta priors fluctuate across runs, reducing answer accuracy despite moderate sampling cost.

In contrast, when $K > 3$, difficulty partitions become increasingly fine-grained and less reliable. The estimated ASCC priors across adjacent difficulty levels become numerically close, while the probability of assigning questions to higher difficulty buckets increases. This leads to overly conservative stopping behavior, resulting in a marked increase in sampling cost and a gradual degradation in answer accuracy due to mis-binning and premature termination on truly hard questions. These observations collectively motivate the choice of $K = 3$ as a robust operating point for ASCC.

### C.3. Classifier Accuracy Regimes and Robustness

Table 5 summarizes the end-to-end performance of TMAFC under different classifiers. LLM-based classifiers achieve substantially lower sampling cost and higher answer accuracy compared to lightweight baselines. In particular, classifiers based on GPT-4 and DeepSeek-V3 closely match the oracle classifier, while GPT-3.5-Turbo incurs only a marginal accuracy drop.

In contrast, lightweight classifiers such as XGBoost and DistilBERT operate far outside this robustness regime. Their low classification accuracy leads to unstable difficulty assignments and ineffective ASCC calibration, resulting in both higher sampling cost and substantially lower answer accuracy. These results indicate that difficulty classification for ASCC requires strong semantic and logical modeling capacity, particularly under cross-domain settings, and that LLM-based classifiers with carefully designed prompts constitute the most effective and practical solution.

*Table 5.* TMAFC performance with different classifiers.

| Classifier | Classification Acc. | Sampling Cost | Answer Acc. |
|---|---|---|---|
| XGBoost | 36.3% | 19.76 | 75.6% (-8.6%) |
| DistilBERT | 42.1% | 19.73 | 78.5% (-6.7%) |
| LLM-based (GPT-3.5-Turbo) | 78.52% | 15.79 | 83.8% (-0.4%) |
| LLM-based (DeepSeek-V3) | 81.3% | 14.88 | 84.2% (-0.0%) |
| LLM-based (GPT-4) | 83.5% | 14.88 | 84.2% (-0.0%) |
| Oracle Classifier | 100% | 14.88 | 84.2% |

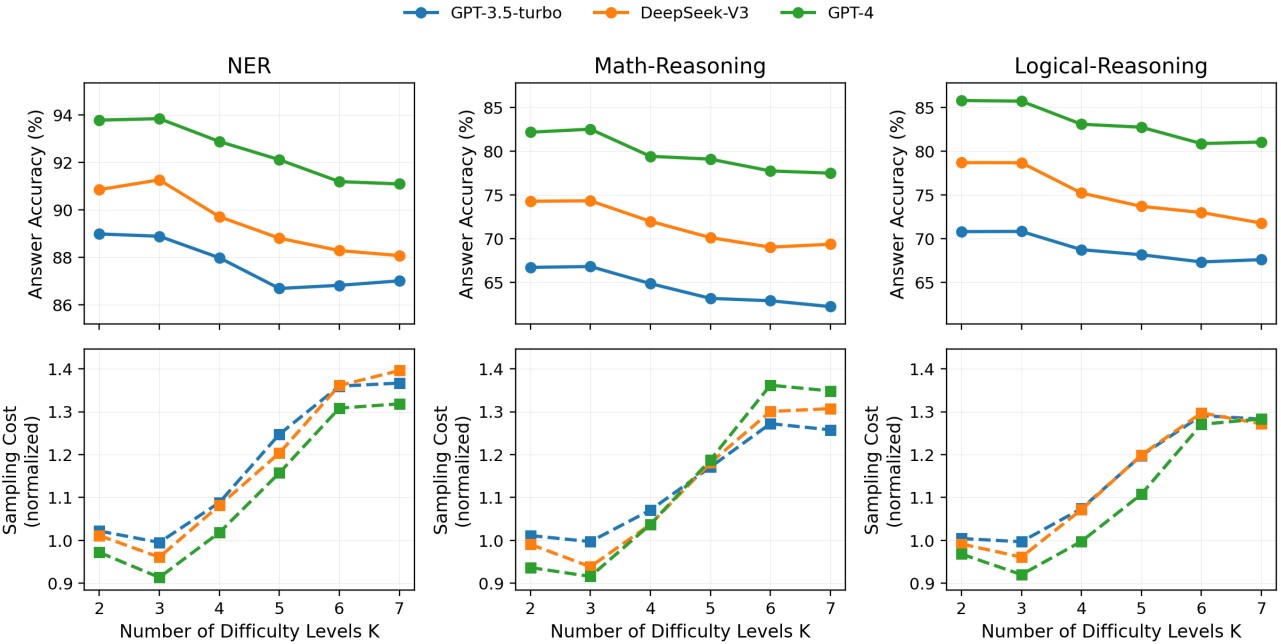

*Figure 11.* **Sweeping the number of difficulty levels $K$ across classifiers and tasks.** We evaluate multiple LLM-based classifiers on NER, mathematical reasoning, and logical reasoning benchmarks. **Top:** end-to-end answer accuracy is maximized at $K=3$ and degrades for both coarser ($K=2$) and finer ($K>3$) difficulty partitions. **Bottom:** sampling cost is minimized at $K=3$; larger $K$ leads to increasingly conservative stopping and higher cost.

