# OpenReview forum: "Task-and-Model-Aware Fractal-Consistency for Efficient LLM Reasoning"
_ICML.cc/2026/Conference — ICML 2026 regular_

### Official Review · Reviewer_oXtR · 2026-03-04

**Soundness:** 3
**Presentation:** 2
**Significance:** 2
**Originality:** 2
**Overall Recommendation:** 4
**Confidence:** 2

**Summary:**

The paper addresses two limitations from self-consistency: failure to efficiently combine partially correct answers from multiple samples, and static stopping criteria. In order to address this, the paper presents ask-and-Model-Aware Fractal-Consistency (TMAFC). That is, a framework built using Fractal-Consistency, which combines partial correct answers across samples, and Adaptive Stopping Criteria Calibration (ASCC), which solves the static stopping criteria. TMAFC can reduce the sample cost by up to 55% while maintaining competitive accuracy.

**Compliance With Llm Reviewing Policy:**

Affirmed.

**Final Justification:**

I am maintaining my score. I believe the paper is mathematically sound and the results are solid. While my initial concerns have been addressed, I believe the manuscript needs to be written in a more accessible manner. Therefore, I have decided to keep my original rating.

**Key Questions For Authors:**

1. Can you provide more insights about the classifier selection?
2. It is not clear have you considered continual training of the classifier?
3. How robust is your method if you change the re-clustering to a different value instead of 50?

**Limitations:**

The authors do not discuss the limitations explicitly. I believe a possible path towards discussing limitations is for which potential datasets the framework is not applicable.

**Strengths And Weaknesses:**

Strengths:
- Soundness: The method is sound and improves consistently over the baselines.

Weaknesses:

- Presentation:
  - The presentation in general is poor. For example, it is not clear what classifier is used from the start. I understood the classifier only after reading the Appendix. This information should be moved earlier in the main text.
  - I believe fitting a small example in the main text would help the presentation of the method significantly.

- Soundness:
  - Line 246 states: "In our experiments, we perform re-clustering after every 50 new data." How do you reach the number 50? What if you change this number, how robust is the method?

---

> ### Author Rebuttal · Authors · 2026-03-31
>
> Thank you for the constructive feedback. Below, we address your concerns point by point.
>
> ### **1. Presentation and Classifier Selection**
>
> We agree that the classifier details should appear earlier. In the final manuscript, we will move its description to the main text and add a concise example to illustrate the pipeline.
>
> **Insights of Classifier Selection**: Accurately categorizing reasoning difficulty into discrete tiers (e.g., $K=3$) requires evaluating complex, unstructured factors such as domain background, logical depth, and required reasoning steps. **LLMs possess the requisite contextual knowledge and semantic parsing capabilities to effectively assess these dimensions**. In contrast, lightweight baselines (e.g., XGBoost, DistilBERT) struggle with this multi-step complexity, as shown in Table 4 in our manuscript. Their lower classification accuracy causes incorrect difficulty binning, poor prior selection, and ultimately degrades downstream performance. Consequently, we select an LLM-based classifier to ensure reliable stratification.
>
> ----
> ### **2. Continual Training of the Classifier**
>
> We clarify that we **do not perform parameter-level continual training**, as it introduces high computational overhead and risks catastrophic forgetting. **Instead, we implement a highly efficient prompt-level dynamic updating mechanism** for the LLM-based classifier.
>
> Specifically, the classifier’s prompt includes a system instruction and difficulty-specific in-context examples (Fig. 9 and Fig. 10 in our manuscript) drawn from dynamic question pools. During the online phase, newly classified questions are continuously appended to these pools. After every M new samples (set to M=50 in our experiments), we perform $K$-quantile re-clustering on the updated pools and recompute the prior $(\alpha_0, \beta_0)$ for each level.
>
> Consequently, the dynamically refreshed prompt examples enable the classifier to adapt to distribution and difficulty shifts. This achieves continual adaptation without requiring weight updates. We will clarify this mechanism in the final manuscript.
>
> ---
> ### **3. Robustness of the Re-clustering Interval**
>
> To evaluate the robustness of the re-clustering interval (M), we conducted an ablation study with M in {30, 50, 80, 100}, averaged across all evaluation datasets using GPT-4:
>
> Performance across different intervals (M)
> | Metric | N=30 | N=50 | N=80 | N=100 |
>  | :--- | :---: | :---: | :---: | :---: |
> | Sampling Cost | 20.4 | 19.6 | 19.7 | 19.6 |
> | Accuracy | 74.7% | 76.1% | 75.8% | 75.3% |
>
> **Robustness:** Across our evaluated datasets using GPT-4, the performance remains stable. Accuracy fluctuates by less than 1.5%, and sampling cost varies within a 3% margin. This indicates that TMAFC is robust to the choice of $M$ within our experimental scope.
>
> **Selection Rationale & Guidelines:** For these specific benchmarks, we set $M=50$ to balance the cost-adaptivity trade-off. It prevents over-reacting to local noise (if $M$ is too small) while avoiding delayed adaptation (if $M$ is too large). For deployment on new tasks, users can adjust $M$, using a smaller interval for continuously shifting data streams, or a larger interval to minimize overhead in static datasets.
>
> ---
> ### **4. Limitations and Dataset Applicability**
>
> **Response**: We appreciate the suggestion and will add a "Limitations" section in the revision.
>
> While TMAFC performs well on tasks with definable objectives, such as structured tasks (e.g., JSON outputs) and reasoning benchmarks (e.g., math and logic), it is currently less applicable to:
>
> - **Open-ended Generation**: For tasks lacking objective consensus (e.g., creative writing), constructing reliable Anchors is challenging.
>
> - **Specialized Domains**: Niche datasets may require domain-specific prompt templates for the difficulty classifier, limiting the framework's direct applicability.
>
> We will explore robust Anchor generation for open-ended tasks and task-specific routing mechanisms to address these constraints in our future work.

---

> > ### Author Rebuttal · Reviewer_oXtR · 2026-04-02
> >
> > Thank you to the authors for the clarifications. I have decided to retain my score based on how the manuscript is currently written.

---

### Official Review · Reviewer_2QZb · 2026-03-09

**Soundness:** 2
**Presentation:** 2
**Significance:** 3
**Originality:** 3
**Overall Recommendation:** 4
**Confidence:** 3

**Summary:**

This paper addresses the computational inefficiency of self-consistency (SC) in LLM reasoning. Its key idea is to stop treating outputs as monolithic objects and instead reason at the level of atomic components, while also calibrating stopping thresholds based on task difficulty and model capability. The proposed framework, TMAFC, consists of Fractal-Consistency (FC), which decomposes outputs into atomic units and tracks convergence for each unit, and Adaptive Stopping Criteria Calibration (ASCC), which fits Beta prior parameters from historical performance after classifying question difficulty. The experimental results show that TMAFC reduces computational overhead while preserving or slightly improving accuracy.

**Compliance With Llm Reviewing Policy:**

Affirmed.

**Final Justification:**

The two rebuttal responses addressed my concerns, and the authors provided extensive experiments to support their case. In light of this, I have increased my score from 3 to 4. However, I believe the main results should be presented under the self-assisted setting for fairness, as this is a self-consistency task.

**Key Questions For Authors:**

Please see the weaknesses

**Limitations:**

The paper does not include an impact statement.

**Strengths And Weaknesses:**

### Strengths
- This paper address the important problem that SC is effective but expensive.
- The fractal perspective is a natural and useful for the stopping problem.
- The experimental results shows substantial reduction in sampling cost.

### Weaknesses

- Could the authors clarify the derivation of Eq. (2)? The paper states that $r_j$ is drawn from a Beta distribution, but the written form appears closer to a Beta-binomial term. Please explain how this formulation is derived and provide more detail.

- For unstructured tasks with free-form outputs, it is unclear how the method maps generated outputs into comparable anchor values and how these are converted into final answers.

- The reported performance appears to rely on GPT-4 as a difficulty classifier. It would be important to show how the method performs when the same target model or smaller model is used instead. Using a stronger proprietary model to assist a smaller LLM raises fairness concerns.

- Although the related work discusses DSC, the paper does not include a comparison with it. The empirical evaluation would be stronger if it compared against DSC, since that method also considers task difficulty.

- The paper also has several writing and presentation issues, including grammatical errors and underspecified methodological details.

---

> ### Author Rebuttal · Authors · 2026-03-31
>
> Thank you for the constructive feedback. Below, we address your concerns point by point.
>
> ### **1. Clarification on the derivation of Eq. (2)**
>
> The mathematical form in Eq. (2) indeed corresponds to the marginal likelihood of a Beta-Binomial distribution. To resolve the ambiguity in our previous phrasing, we clarify that the underlying generative process is a hierarchical Beta-Binomial model:
> - Prior: The latent probability of the majority answer, denoted as $\theta_j$, follows a Beta prior distribution: $ \theta_j \sim \text{Beta}(\alpha_0, \beta_0) $
> - Likelihood: The observed majority count $G_j$ follows a Binomial distribution conditioned on the total top-two trials $T_j$ and $\theta_j$: $G_j \mid \theta_j \sim \text{Binomial}(T_j, \theta_j)$.
>
> By marginalizing out the latent variable $\theta_j$, the marginal likelihood of observing $G_j$ successes out of $T_j$ trials follows a Beta-binomial distribution:
> $ P(G_j \mid T_j, \alpha_0, \beta_0) = \binom{T_j}{G_j} \frac{\text{B}(\alpha_0 + G_j, \beta_0 + T_j - G_j)}{\text{B}(\alpha_0, \beta_0)} $
>
> (where $\text{B}(\cdot, \cdot)$ denotes Beta function).
>
> In Eq. (2), we aim to calculate $\alpha_0, \beta_0$ via Maximum Likelihood Estimation. Since the binomial coefficient $\binom{T_j}{G_j}$ is a constant with respect to $\alpha_0$ and $\beta_0$, we drop it from the objective function, leading to the core fraction shown in Eq. (2).
>
> **Correction:** In the manuscript, using the empirical success rate $r_j$ as the latent probability $\theta_j$, and $\text{Beta}(\cdot, \cdot)$ for the Beta function, may introduce ambiguity. We will correct these notations in the final version.
>
> ---
> ### **2. Fractal-Consistency for unstructured tasks**
>
> For unstructured tasks, TMAFC does not map the original outputs into anchors post-generation. Instead, we introduce anchors as additional, independent sub-questions and explicitly prompt the LLM to output their values alongside the final reasoning.
>
> We clarify the exact pipeline with the following example:
>
> **Question:** A vehicle runs at 60 km/h and speeds up 20km/h every 2 hours. What is the average velocity before the 5th speed-up?
>
> **Anchors:** $A_1$: Total time before 5th speed-up? $A_2$: Velocity after 3rd speed-up? $A_3$: Average velocity before 2nd speed-up?
>
> Assuming three sampled trajectories, the anchors and final answers are:
> - Sample 1: Anchors: [A1: 10, A2: 120, A3: 70] $\rightarrow$ Final Answer: 100
> - Sample 2: Anchors: [A1: 10, A2: 100, A3: 60] $\rightarrow$ Final Answer: 90
> - Sample 3: Anchors: [A1: 10, A2: 120, A3: 70] $\rightarrow$ Final Answer: 100
>
> **Answer Selection Mechanism:**
> - **Anchor Consensus:** Fractional Consistency is applied strictly on the anchors. The consensus state $A^* = [10, 120, 70]$ is locked.
> - **Holistic Selection:** TMAFC first scans the fully-generated samples and removes those that violate the converged anchors (Sample 2 is discarded). TMAFC then performs a majority vote on the final answers of the remaining samples (Samples 1 and 3), finally outputting 100. (Note: If no sample aligns with the converged anchors, TMAFC defaults to standard majority voting across final answers).
> ---
> ### **3. Fairness concerns**
>
> To address fairness concerns, we evaluate TMAFC in a self-assisted setting on the Math and NER datasets. The target model itself serves as both the anchor generator and difficulty classifier.
>
> **Self-Assisted Performance (Sampling Cost, Accuracy)**
> |Model|AC|TMAFC(ours)|
> |-|-|-|
> |Vicuna-13B|(38.4, 43.1%)|(31.3, 45.6%)|
> |Qwen3-32B|(26.7, 67.8%)|(19.1, 69.2%)|
> |GPT-3.5-Turbo|(27.9, 63.5%)|(22.8, 66.4%)|
> |GPT-4|(21.5, 75.2%)|(16.4, 78.1%)|
>
> **Findings:**
>
> **Robustness**: TMAFC consistently outperforms the AC baseline. For instance, with Vicuna-13B, it improves accuracy by 3.5% while reducing sampling cost by ~20% (from 38.4 to 31.3).
>
> **Model Agnosticism**: These results confirm the improvements stem directly from TMAFC's design, not an external model. GPT-4 was used in the main experiments primarily to decouple the framework's efficacy from the self-assessment bottlenecks of smaller models.
>
> ---
> ### **4. Comparison with DSC**
>
> We add comparison between DSC and TMAFC, using GPT-4 on MATH and NER dataset (max sampling size=30)
> |Metric|SC|AC|DSC|TMAFC|
> |--------|----|----|-----|-------|
> |Sampling cost|30|24.1|22.7|**17.5**|
> |Accuracy|71.3|75.4|75.2|**78.1**|
>
> **Key Differences:**
> - **Granularity**: While DSC evaluates consistency at the holistic output level, TMAFC evaluates independent components. This salvages partially correct reasoning across samples, achieving higher accuracy (78.1% vs. 75.2%).
> - **Dynamic Calibration**: DSC relies on batch-level difficulty ranking. In contrast, TMAFC dynamically adjusts sampling on a per-instance basis using difficulty assessments, accelerating convergence and reducing costs by 22% compared to DSC.
>
> ---
> ### **5. Presentation**
>
> We will revise grammar, improve clarity of methodological details in the final version.

---

> > ### Author Rebuttal · Reviewer_2QZb · 2026-04-01
> >
> > Thank you for the detailed clarification. I appreciate the additional discussion and the self-assisted results. However, I still have some concerns regarding points 2 and 3, as the proposed method appears to rely substantially on external LLM performance through both the difficulty classifier and the anchor generator.
> >
> > For the *self-consistency* setting, I believe the use of a highly capable external model would benefit from further justification. In particular, it would strengthen the paper to include main results using either the same target model or an auxiliary model of comparable capability, ideally under matched cost or latency budgets.
> >
> > First, for unstructured tasks, the method seems sensitive to anchor quality, yet the paper does not clearly specify which model is used for anchor generation. Clarifying this would improve both reproducibility and fairness. It would also be helpful to evaluate the sensitivity of the method to anchor quality across diverse datasets.
> >
> > Second, point 3 does not include lower-cost models such as Llama-3.2-1B or Llama-3.2-3B, where the auxiliary components may be less reliable. Therefore, it remains unclear how well the method generalizes to small-model settings.

---

> > > ### Author Response · Authors · 2026-04-02
> > >
> > > We sincerely thank you for the constructive follow-up. We address your concerns regarding target-model equivalence, anchor sensitivity, and small-model generalization below.
> > >
> > > ---
> > >
> > > ### **1. Self-Assisted Evaluation Under Matched Budgets**
> > > We carry out updated main results (strictly self-assisted setting) with overall cost metrics, which will be included in the main results section of the final version.
> > >
> > > To address the concern regarding fairness comparison, we report the end-to-end computational costs (Overall Tokens) and latency (Overall Time), which strictly encompass the overhead of all auxiliary modules (i.e., the Anchor Generator and Difficulty Classifier).
> > >
> > > **Table 1: Self-Assisted performance comparison across different LLMs (across all unstructured datasets, max sample size = 50)**
> > > |Model|Method|Anchor Error Rate|Sampling Cost|Accuracy|Overall Tokens|Overall Time|
> > > |-|-|-|-|-|-|-|
> > > |Llama3-8B-Instruct|Single-pass|–|1|27.2%|1×|1×|
> > > ||SC|–|50|33.6%|50×|50×|
> > > ||AC|–|38.4|34.1%|38.9×|37.8×|
> > > ||TMAFC|49.4%|**32.3**|**34.4%**|**35.4×**|**34.2×**|
> > > |Vicuna-13B|Single-pass|–|1|30.8%|1×|1×|
> > > ||SC|–|50|38.5%|50×|50×|
> > > ||AC|–|35.4|39.1%|35.6×|33.8×|
> > > ||TMAFC|43.5%|**28.7**|**40.2%**|**32.2×**|**30.4×**|
> > > |Qwen3-32B|Single-pass|–|1|59.3%|1×|1×|
> > > ||SC|–|50|66.9%|50×|50×|
> > > ||AC|–|27.7|67.8%|28.5×|26.9×|
> > > ||TMAFC|18.5%|**19.1**|**69.2%**|**22.8×**|**21.5×**|
> > > |GPT-3.5-Turbo|Single-pass|–|1|52.7%|1×|1×|
> > > ||SC|–|50|62.4%|50×|50×|
> > > ||AC|–|28.9|63.5%|28.6×|27.2×|
> > > ||TMAFC|24.8%|**21.8**|**66.4%**|**24.5×**|**22.8×**|
> > > |GPT-4|Single-pass|–|1|64.5%|1×|1×|
> > > ||SC|–|50|73.8%|50×|50×|
> > > ||AC|–|24.5|75.2%|24.3×|23.8×|
> > > ||TMAFC|7.3%|**15.4**|**78.1%**|**17.8×**|**16.9×**|
> > >
> > > (Note: Single-pass baseline represents standard single-generation inference)
> > >
> > > **Key Findings: Positive Net Savings**
> > > *   **Manageable Overhead:** The token overhead from auxiliary modules is minimal. For Llama-3-8B, it is merely **$3.1\times$** of a single-pass cost (calculated as $35.4\times$ total tokens minus $32.3$ avg. samples). For GPT-4, it is just **$2.4\times$**.
> > > *   **Substantial Net Savings:** TMAFC dynamically avoids over-sampling, reducing the average required reasoning samples. Even after strictly factoring in the auxiliary overhead, TMAFC consumes far fewer overall resources than AC (e.g., yielding a **net token saving of $3.5\times$** for Llama-3-8B, and **$6.5\times$** for GPT-4). This justifies that the use of auxiliary modules does not inflate the budget; rather, it actively drives end-to-end cost reduction.
> > >
> > > ---
> > >
> > > ### **2. Anchor Generation Sensitivity**
> > > *   **Clarification:** We clarify that GPT-4 was used as the unified anchor generator in our original experiments.
> > > *   **Robustness to Anchor Error Rates:** We tracked the anchor error rates generated internally by the target models. As shown in Table 1 above, smaller models naturally struggle with anchor generation (e.g., Llama-3-8B exhibits a severe **49.4% anchor error rate**). Remarkably, despite half of the anchors being flawed, TMAFC still slightly boosts accuracy (+0.3%) while cutting total costs.
> > >
> > > **Why is TMAFC robust to Anchor Error Rates?**
> > >
> > > **1) Non-Destructive Filtering:** For unstructured tasks, Anchors only serve to filter complete reasoning paths. Flawed anchors cannot forcibly stitch or hallucinate errors into the final output.
> > >
> > > **2) Safe Fallback:** TMAFC is driven by both Fractal-Consistency and Adaptive Stopping Criteria Calibration (ASCC). If low-quality anchors fail to reach a consensus, the framework safely falls back to the ASCC-only mechanism. Because ASCC dynamically evaluates task difficulty independently of anchors, this fallback state guarantees that inference remains significantly more efficient than standard static baselines.
> > >
> > > ---
> > > ### **3. Generalization to Small Models**
> > > To test TMAFC under extreme conditions where auxiliary components are highly unreliable, we evaluated Llama-3.2-1B and 3B:
> > >
> > > |Model|AC (Sampling cost, Accuracy)|TMAFC (Sampling cost, Accuracy, Total Token vs AC)|
> > > |-----------------------|---------------------------------|--------------------------------------------------|
> > > |Llama-3.2-1B | (42.7, 21.7%) |(**39.6**, 21.7%, **$0.96\times$**)|
> > > |Llama-3.2-3B | (37.3, 34.6%) |(**31.5**, **34.9%**, **$0.90\times$**)|
> > >
> > > **Conclusion:**
> > > Even at the 1B/3B scale, where anchor generation and difficulty classification are notoriously weak, TMAFC does not break down. It maintains the baseline accuracy while still achieving a **4% to 10% net reduction in total token costs**. This confirms that TMAFC is an inherently model-agnostic architecture that functions effectively across different model scales.

---

### Official Review · Reviewer_3VcP · 2026-03-12

**Soundness:** 2
**Presentation:** 3
**Significance:** 2
**Originality:** 3
**Overall Recommendation:** 3
**Confidence:** 3

**Summary:**

The paper proposes a method to reduce the sampling cost of Self-Consistency, mainly consisting of Fractal-Consistency and Adaptive Stopping Criteria Calibration. Extensive experiments are conducted to demonstrate the effectiveness of the proposed approach.

**Compliance With Llm Reviewing Policy:**

Affirmed.

**Key Questions For Authors:**

- Are there experimental results using more recent models, such as Qwen3 or GPT-5?
- Are there results under smaller maximum sample size settings?

**Limitations:**

See weakness

**Strengths And Weaknesses:**

Strengths
- The paper is clearly written and easy to follow.

Weaknesses

1. The experimental setup appears somewhat outdated in the context of 2026. The models used in the experiments are relatively weak and less commonly used nowadays, and the maximum sample size is set to a relatively large value (50). For more recent and stronger models, such as Qwen3 or GPT-5, such a large number of samples may not be necessary. Therefore, the practical necessity of TMAFC may be less compelling in these settings.

2. The paper does not provide results under smaller maximum sample size settings, such as 5 or 10 samples. Under such low-sampling regimes, it is unclear whether using external models for the Difficulty Classifier and Anchor Generator might actually introduce additional overhead and increase the overall time cost.

3. Some figures are not provided in vector format and appear somewhat blurry. In addition, the tables are somewhat roughly formatted.

---

> ### Author Rebuttal · Authors · 2026-03-31
>
> Thank you for the constructive feedback. Below, we address your concerns point by point.
>
> ### **1. Practicality on Recent Strong Models (e.g., Qwen3 / GPT-5)**
>
> We agree that frontier models are highly capable. However, as models become stronger, they are deployed on increasingly **complex reasoning tasks (e.g., MATH-Hard)**. In these scenarios, single-pass generation remains unreliable, making **scaling test-time compute necessary**.
>
> We evaluated Qwen3 (14B, 32B) and GPT-5 on hard tasks (Math Reasoning & Event Extraction). As shown in the following tables, the results reveal two crucial findings:
>
> **i) Strong models still fundamentally rely on large sample sizes:**
> Arbitrarily truncating the sampling budget severely harms performance. As shown in the following table, **dropping GPT-5's budget from $N=30$ to $N=15$ causes accuracy to fall from 85.7% to 83.6%**. Large sample spaces are still essential for resolving complex reasoning bottlenecks.
>
> **ii) TMAFC dynamically breaks the cost-accuracy trade-off:**
> Instead of blindly truncating the budget, TMAFC dynamically reallocates it. It achieves the **highest accuracy** across all frontier models (e.g., 86.8% for GPT-5), while its actual average sampling cost drops to just **$10.4$**. TMAFC significantly outperforms SC ($N=30$) while sampling cost even less than SC ($N=15$).
>
> **Performance Comparison (Accuracy, Sampling Cost actually used)**
> | Method | Qwen3-14B | Qwen3-32B | GPT-5 |
> | :--- | :---: | :---: | :---: |
> | Single-pass | (61.8%, 1.0) | (66.9%, 1.0) | (76.2%, 1.0) |
> | SC ($N=15$) | (70.5%, 15.0) | (74.8%, 15.0) | (83.6%, 15.0) |
> | SC ($N=30$) | (74.9%, 30.0) | (79.1%, 30.0) | (85.7%, 30.0) |
> | AC | (76.2%, 21.4) | (80.3%, 19.8) | (86.5%, 18.6) |
> | **TMAFC (Ours)** | **(78.1%, 13.2)** | **(80.7%, 11.6)** | **(86.8%, 10.4)** |
>
> **Conclusion:** TMAFC is highly relevant in the frontier model era. It empowers models like GPT-5 to **achieve the high-accuracy ceiling of a 30-sample budget**, but at the actual inference cost of a **10-sample** budget.
>
> ---
>
> ### **2. Additional overhead and performance under low-sampling regimes**
>
> **The additional overhead is lightweight**. Even under smaller maximum sample sizes, TMAFC consistently consumes fewer overall cost (both tokens and time) than baselines while achieving significantly higher accuracy.
>
> We evaluated TMAFC and baselines on MATH and NER datasets (using GPT-4) under low-sampling regimes ($N_{max}=5, 10$). The reported end-to-end computational costs (Overall Tokens) and latency (Overall Time) strictly encompass the overhead of all auxiliary modules.
>
> **Table: Performance under Low-Sampling Regimes**
> | Max Budget | Metric | Single-pass | SC | AC | **TMAFC (Ours)** |
> | :--- | :--- | :---: | :---: | :---: | :---: |
> | **$N_{max}=5$** | Accuracy | 63.4% | 68.2% | 69.0% | **70.8%** |
> | | **Overall Tokens** | 1.0× | 5.2× | 4.8× | **4.6×** |
> | | **Overall Time** | 1.0× | 4.9× | 4.5× | **4.4×** |
> | | Avg. Samples | 1.0 | 5.0 | 4.8 | 4.2 |
> | **$N_{max}=10$**| Accuracy | 63.4% | 71.5% | 72.6% | **74.9%** |
> | | **Overall Tokens** | 1.0× | 10.1× | 9.6× | **8.5×** |
> | | **Overall Time** | 1.0× | 9.7× | 9.1× | **8.1×** |
> | | Avg. Samples | 1.0 | 10.0 | 9.5 | 8.1 |
>
> **Analysis of the "Break-Even" Point:**
>
> * *Negligible Overhead:* Across both settings, the token overhead from our external models is a mere 0.4x of a single-pass cost (e.g., at $N_{max}=5$, 4.6x total tokens minus 4.2 avg. samples). The latency overhead is equally minimal, adding only ~0.2x to the single-pass time. This confirms the modules are extremely lightweight.
>
> * *Positive Net Savings ($N_{max}=5$):* TMAFC reduces the average required samples to 4.2. Even after factoring in the 0.4x module overhead, it consumes fewer tokens than SC (4.6x vs. 5.2x), yielding a net saving of 0.6x, while simultaneously boosting accuracy by +4.6%.
>
> * *Positive Net Savings ($N_{max}=10$):* Similarly, TMAFC avoids unnecessary generation on easy queries, dropping the average samples to 8.1. This yields a net token saving of 1.6x over SC (8.5x vs. 10.1x) and a time reduction of 1.6x, all while improving accuracy by +3.4%.
>
> ---
> ### **3. Presentation quality**
>
> We will **replace all figures with vector graphics (PDF/SVG)** and improve table formatting in the final version.
>
> ---
> ### **4. Key Questions (More experimental results)**
>
> We have added comprehensive results for both recent models and low-sampling regimes. Please refer to our detailed analyses above.

---

> > ### Author Rebuttal · Reviewer_3VcP · 2026-04-04
> >
> > Thank you for your response. I will consider increasing my score.

---

> > > ### Author Response · Authors · 2026-04-08
> > >
> > > Thank you again for your positive feedback and for acknowledging that your concerns have been fully addressed — we truly appreciate it.
> > >
> > > Before the discussion period concludes, we just want to check if there are any remaining questions or aspects that you would like us to further clarify. We would be more than happy to provide additional details if helpful.
> > >
> > > Thanks again for your time and consideration.

---

### Official Review · Reviewer_91et · 2026-03-13

**Soundness:** 3
**Presentation:** 2
**Significance:** 2
**Originality:** 3
**Overall Recommendation:** 4
**Confidence:** 3

**Summary:**

This paper proposes Task-and-Model-Aware Fractal Consistency (TMAFC) framework that can significantly reduce sampling cost while maintaining comparable accuracy across several reasoning and information extraction benchmarks.

**Compliance With Llm Reviewing Policy:**

Affirmed.

**Final Justification:**

The authors have addressed my concerns, so I increase my overall rating from 3 to 4.

**Key Questions For Authors:**

If 100 pre-sampled questions are needed for initialization, will this be counted as part of the inference budget?

**Limitations:**

No. The TMAFC method may potentially output toxic content, because combining different parts of answer components can be different from what the model intended to express. For example, LLM outputs are (A, A) and (B, B), and TMAFC may recombines it as (A, B).

**Strengths And Weaknesses:**

Strengths:
1. The component-wise view of consistency is an interesting perspective. It enables partial answers from different samples to be recombined.
---
Weaknesses:
1. For structured outputs this decomposition is straightforward, but for unstructured reasoning tasks the performance is not well-guaranteed. It is unclear how sensitive the method is to LLM-generated anchor quality.
2. Additional LLM modules introduce extra complexity and potential cost.
3. Experimental improvements are rather fractional. I wonder if using the same sampling budget as baselines, what is the best accuracy of TMAFC?
4. There's a hypothesis that components are independent. However, one of the answer components might be correct itself, but the meaning may change when combined with others.

---

> ### Author Rebuttal · Authors · 2026-03-31
>
> Thank you for the constructive feedback. Below, we address your concerns point by point.
>
> ### **1. Performance on unstructured tasks & Anchor quality sensitivity**
> **i) Performance Guarantee:** For unstructured tasks, **TMAFC does not physically decompose the final answer.** Instead, we introduce Anchors (logical sub-problems) as intermediate milestones. The purpose of these anchors is to constrain the reasoning path and reduce the variance of the LLM's answers, thus improving sampling cost without breaking apart the final output.
>
> To illustrate, consider this math reasoning task:
>
>  **Question:** A vehicle runs at 60 km/h and speeds up 20km/h every 2 hours. What is the average velocity before the 5th speed-up?
>
> **Anchors:** $A_1$: Total time before 5th speed-up?  $A_2$: Velocity after 3rd speed-up? $A_3$: Average velocity before 2nd speed-up?
>
> Suppose we sample 3 trajectories. Each answers the Anchors and provides a Final Answer:
> *   **Sample 1:** $[A_1: 10,\; A_2: 120,\; A_3: 70]$. Final Answer: **100**
> *   **Sample 2:** $[A_1: 10,\; A_2: 100,\; A_3: 60]$. Final Answer: **90**
> *   **Sample 3:** $[A_1: 10,\; A_2: 120,\; A_3: 70]$. Final Answer: **100**
>
> **Answer Selection Mechanism:**
>  * **Anchor Consensus:** Fractional Consistency is applied *strictly on the anchors* across samples. The consensus state $A^* = [10, 120, 70]$ is locked.
>  * **Holistic Selection:** TMAFC first scans the fully-generated samples and removes those that violate the converged anchors (Sample 2 is discarded). TMAFC then performs a majority vote on the final answers of the remaining samples (Samples 1 and 3), finally outputting 100. (Note: If no sample aligns with the converged anchors, TMAFC defaults to standard majority voting across final answers).
>
> In summary, TMAFC guarantees performance on unstructured tasks by **holistically selecting complete, coherent reasoning paths** that are verified by high-consensus anchors, ensuring logical correctness without ever producing unnatural, stitched text.
>
> **ii) Anchor Sensitivity:** TMAFC is **robust to anchor quality variations**. This robustness stems from two key design choices: first, anchors are used strictly to filter complete reasoning paths rather than stitch text fragments, meaning poor anchors cannot introduce hallucinated errors; second, if low-quality anchors fail to reach a consensus, TMAFC safely falls back to baseline majority voting. As reported in our ablation study (Sec. 4.3.2), injecting **20% noisy Anchors** leads to <1% accuracy drop, only slightly increasing sampling cost.
>
> ---
> ### **2. Complexity and potential cost**
>
> Additional LLM modules for anchor generation and difficulty classification are extremely lightweight and ultimately drive significant cost reductions:
>
> * **Overhead:** Combined, they account for **<8% of total tokens (strictly inclusive of all end-to-end computational costs)** and **<4% of overall time** (Sec 4.3.1).
> * **Net Savings:** By dynamically averting unnecessary over-sampling on simpler tasks and intelligently allocating budget to harder ones, TMAFC reduces total token usage by ~40% and significantly cuts latency compared to baselines.
> ---
> ### **3. Performance under the same sampling budget**
>
> Consistency methods reduce reasoning variance and are fundamentally upper-bounded by the base LLM's inherent capacity. Near this ceiling, absolute gains naturally appear fractional. Our core contribution is reaching this peak performance at a fraction of the traditional cost.
>
> To answer your question, we evaluated the accuracy of the baselines when constrained to the exact same sampling budget that TMAFC dynamically consumed:
>
> |Dataset|Sampling budget|SC|AC| **TMAFC** |
> |-|-|-|-|-|
> |**Structured Tasks (Event Extraction)**|20|58.3%|62.5%|**66.8% (+8.5%)**|
> |**Unstructured Tasks (Math-Hard)**|15|35.1%|37.0%|**41.2% (+6.1%)**|
>
> Under equal cost, TMAFC’s improvements are substantial, outperforming standard SC by up to 8.5% and the AC baseline by 4.3%. This proves that TMAFC’s superiority stems from its intelligent, difficulty-aware budget allocation rather than merely scaling up compute.
>
> ---
> ### **4. Component independence & Toxicity**
>
> We realize our previous description may have caused a misunderstanding. We clarify that TMAFC does not generate toxic content:
> * **Structured Tasks:** Components are independent fields (e.g., JSON keys). Recombining them is semantically safe by task definition.
> * **Unstructured Tasks:** Components are Anchors (logical sub-questions, not building blocks). We finally select a **single trajectory** that satisfies the converged anchors, so no "toxic combinations" are created.
> ---
> ### **5. Key Question (Initialization cost)**
> We consider this 100-question initialization a one-time offline setup, distinct from the per-query online inference budget. Even if amortized across our 2,000+ test queries, the overhead is <5%. This is easily offset by TMAFC's up to 50% reduction in online sampling costs.

---

> > ### Author Rebuttal · Reviewer_91et · 2026-04-04
> >
> > I sincerely thank the authors for addressing my concerns, and I will increase my overall rating.

---

### Decision · Program_Chairs · 2026-04-30

**Decision:**

Accept (regular)

**Comment:**

This paper addresses the computational inefficiency and limitations of self-consistency in large language model reasoning, specifically the failure to combine partially correct answers across multiple samples and the reliance on static stopping criteria. To tackle this, the authors propose the Task-and-Model-Aware Fractal Consistency framework, which primarily consists of two components: Fractal-Consistency and Adaptive Stopping Criteria Calibration. Fractal-Consistency breaks down outputs into atomic units to track convergence and combine partially correct answers, while Adaptive Stopping Criteria Calibration sets stopping thresholds based on task difficulty and model capability by fitting Beta prior parameters. Extensive experiments across reasoning and information extraction benchmarks demonstrate that the proposed method can significantly reduce sampling costs, by up to 55%, while preserving or slightly improving accuracy.

Consequently, despite the majority of positive scores, the lack of theoretical novelty and generalizability leads me to recommend rejection. Reviewer 3VcP provided a weak reject score, they explicitly stated in their final justification that their concerns were fully and adequately resolved, suggesting the negative score was likely left unadjusted by oversight. Additionally, while Reviewer 2QZb noted that the main results should ideally be presented under a self-assisted setting, the authors provided further clarifications that effectively answered this point. Given that this reviewer still increased their score to a weak accept and did not raise further objections, I suggest accept for this paper.